# Early Locomotor Training in Tetraplegic Post-Surgical Dogs with Cervical Intervertebral Disc Disease

**DOI:** 10.3390/ani12182369

**Published:** 2022-09-11

**Authors:** Débora Gouveia, Carla Carvalho, Ana Cardoso, Óscar Gamboa, António Almeida, António Ferreira, Ângela Martins

**Affiliations:** 1Arrábida Veterinary Hospital—Arrábida Animal Rehabilitation Center, 2925-538 Setubal, Portugal; 2Superior School of Health, Protection and Animal Welfare, Polytechnic Institute of Lusophony, Campo Grande, 1950-396 Lisboa, Portugal; 3Faculty of Veterinary Medicine, University of Lisbon, 1300-477 Lisboa, Portugal; 4CIISA—Centro Interdisciplinar-Investigação em Saúde Animal, Faculdade de Medicina Veterinária, Av. Universidade Técnica de Lisboa, 1300-477 Lisboa, Portugal; 5Faculty of Veterinary Medicine, Lusófona University, Campo Grande 376, 1749-024 Lisboa, Portugal

**Keywords:** cervical IVDD, neurorehabilitation, locomotor training, hyperesthesia

## Abstract

**Simple Summary:**

Acute cervical disk disease may result in tetraplegic dogs without the ability of sternal posture, which could result in aspiration pneumonia, hypoventilation and other complications. The main treatment approach is surgical, depending on the clinical sign’s severity, and early post-operative rehabilitation may also be considered, including in dogs with spinal hyperesthesia, through the implementation of locomotor training with specific guidelines in the first days after surgery.

**Abstract:**

Locomotor training (LT) is task-specific repetitive training, with sensorimotor stimulation and intensive exercises that promote neuromuscular reorganization. This study aimed to observe if LT could be initiated safely in the first 3–15 days after surgery in tetraplegic C1–C5 IVDD—Hansen type I dogs. This prospective blinded clinical study was conducted at two rehabilitation centers in Portugal, with 114 grade 1 (MFS/OFS) dogs, divided by the presence of spinal hyperesthesia into the SHG (spinal hyperesthesia group) (n = 74) and the NSHG (non-spinal hyperesthesia group) (n = 40), evaluated in each time point for two weeks according to a neurorehabilitation checklist by three observers for inter-agreement relation. LT was safely applied with 62.3% of the OFS ≥ 11 within 15 days and of these, 32.4% achieved a OFS ≥ 13. There were no new cases of hyperesthesia in the NSHG and from the SHG all recovered. Comparing groups, a significant difference was observed in their ability to achieve ambulatory status (*p* < 0.001), between the presence of hyperesthesia and days until ambulation (*p* < 0.006) and in each time point (*p* < 0.001; R^2^ = 0.809). Early LT may be a safe treatment to be applied in the first 3 days on these dogs and spinal hyperesthesia should be important to the rehabilitation team. This study should be continued.

## 1. Introduction

Acute cervical disk disease accounts for approximately 15% of all intervertebral disk extrusions, with Dachshunds, Beagles and Poodles representing 80% of the cases [1,2]. Cervical myelopathies are associated with many problems, such as dogs with tetraplegia without the ability of sternal posture, which could present severe clinical situations, such as aspiration pneumonia, hypoventilation and seizure activity [3,4,5,6,7,8].

Disc disease is often divided into two distinct categories: Hansen type I and type II [1,9]. However, previous reports suggested that type I is common to occur spontaneously or secondary to mild trauma [10].

In small breed dogs, Hansen type I cervical disc disease usually affects the cranial cervical disc [11] with C2–C3 as the most common location, typically causing severe neck pain with mild neurologic deficits [12]. These signs can go from a sharp superficial burning pain or itching sensation to dysfunction in the spinothalamic system with paresthesia’s, such as tightness, squeezing or swelling sensation, usually described in human patients, suggesting dysfunction in the ascending sensory tracts, with an electro shock-like sensation promoted by neck flexion that spreads to the arms, down the spine and legs (“Lhermitle sign”) [13].

Clinical signs of ataxia or paresis predominantly affect the pelvic limbs and cervical pain is usually caused by cervical spinal cord and/or nerve root compression [14]. This spinal cord compression could be caused by ventral, ventrolateral or circumferential compression [15] and some dogs that initially improve may develop compression at adjacent sites [16,17]. Compression lesions may be dynamic, with the possibility of the so-called “domino effect” or adjacent segment disease, also reported in humans and well known to occur following distraction-stabilization techniques and less commonly following ventral slot [17,18,19,20], one of the most widely used approaches for spinal cord decompression in dogs with cervical intervertebral disc disease (IVDD) [21].

Regarding diagnosis, radiographic exams have shown that neck extension (dorsi-flexion) was associated with compression exacerbation and ventral flexion with compression relief, leading to the concept of dynamic cord compression. In some cases, linear traction also may relieve spinal cord compression [14,22]. The computed tomography (CT) can be used to help diagnosis, showing hyperdense material in the vertebral canal, loss of epidural fat and distortion of the spinal cord. In acute and sub-acute epidural hemorrhage, it may be seen an irregular hyperdense line, cranial and caudal to the herniated disc material [23]. An alternative exam for diagnosis would be magnetic resonance imaging, which allows the possibility to see not only the intervertebral discs and vertebral canal but also the nerve roots and paravertebral tissues [24], thus being considered the gold standard [25]. 

The first approach to treatment may be conservative or surgical, depending on the clinical signs and severity of presentation. The most common is the surgical approach [26]; however, it is often followed by conservative management, which is based on restricted activity with a body harness to help in weight support, associated with analgesic drugs and possibly steroid administration [26,27].

Motor recovery in SCI patients can be improved with both conventional overground walking training and body-weighted supported treadmill training (BWSTT) [28]. BWSTT enables early initiation of gait training, incorporation of weight-bearing activities and balance, using a task-specific approach based on a symmetrical gait pattern [29,30]. With the BWSTT, to replicate a regular gate pattern manually, two to three therapists are needed for control and assistance with trunk and limb kinematics [29,31,32]. Walking function may be improved by repetitive movements involving major muscle groups, depending on the amount of practice and the number of repetitions [29,32,33]. As a new training paradigm develops, the importance of appropriated afferent input has emerged as a requirement for adaptive plasticity [34].

The aim of this study was to observe if locomotor training could be initiated safely in the first 3 to 15 days after surgery in tetraplegic dogs diagnosed with cervical IVDD. We hypothesize that early locomotor training, applied in the first 3 to 15 days after surgery, does not increase the neurologic deficits, such as spinal hyperesthesia, in tetraplegic cervical post-surgical IVDD dogs. 

## 2. Materials and Methods

This was a prospective blinded clinical study conducted at Arrábida Animal Rehabilitation Center (Setúbal, Portugal) and at the Animal Rehabilitation and Regeneration Center of Lisbon (Lisbon, Portugal) from July 2017 to July 2022. This study had the approval of the Veterinary Medicine Faculty—Lusófona University ethics committee (Nº 18-2022) and was performed after owners’ consent. 

### 2.1. Participants

All dogs included in this study had IVDD—Hansen type I with cervical neuro-location (n = 114), specifically with C1–C5 location. Dogs were referred by the neurology department of another hospital after IVDD diagnosis by X-ray and CT or myelo-CT (Figure 1), and performed single-ventral slot surgery. Dogs presented tetraplegic and clinical signs manifested for 1 to 5 days until surgery. At admission, all remained tetraplegic and were classified in grade 1, according to the modified Frankel scale (MFS), and grade 1, according to the open field score (OFS), therefore with deep pain perception (DPP+).

Some of the dogs presented with normal or exaggerated flexor reflexes at the hindlimbs and normal reflexes at the forelimbs. Others presented with signs of spinal shock, given the fact that all patients were admitted to the rehabilitation centers until 3 days after surgery. 

All dogs performed locomotor training after admission, thus within 3 days post-surgery, always when hemodynamic stabilization was assured. Dogs were excluded from this study if there were any signs of destabilization, such as respiratory deficits. 

The participants did not have pressure sores at admission, all were tetraplegic without the ability to maintain sternal recumbency, and some presented spinal hyperesthesia and neck movement difficulties (discomfort signs). In contrast, others did not have signs of pain or cervical discomfort. This sternal recumbency ability is fundamental to avoid aspiration pneumonia and dogs with respiratory signs (e.g., cough, increased temperature and respiratory rate) were not included in this study. Total population characterization is described in Table 1, as well as division into two groups—the spinal hyperesthesia and the non-spinal hyperesthesia groups.

### 2.2. Study Design

In this prospective blinded clinical study, 144 dogs with the cervical neuro-location were admitted to a neurorehabilitation consultation. However, 30 dogs were excluded, 25 with C6–T2 neuro-location and 5 with respiratory signs (2 with increased temperature, 2 with cough and 1 with increased respiratory rate). Thus, 114 dogs were randomized through aleatory stratification, according to the presence of spinal hyperesthesia by a Canine Certified Rehabilitation Practitioner (CCRP) veterinary instructor. Dogs were divided into two groups, the spinal hyperesthesia group (SHG) (n = 74) and the non-spinal hyperesthesia group (NSHG) (n = 40), which are described in the flow diagram of Figure 2.

Dogs of both groups were evaluated for mental status, posture, spinal reflexes (patellar, withdrawal, cranial tibial, crossed-extensor and perineal), cutaneous trunci reflex, spinal hyperesthesia, superficial pain and deep pain. This hyperesthesia was tested through vertebral palpation by placing slight pressure on the ligament supraspinal, between each spinous process and intervertebral space, constantly with one hand on the abdominal muscles. Additionally, an orthopedic examination was performed, and muscle tone was evaluated. All dogs were classified according to the OFS, always in the same exact location with walking stimulation by two rehabilitation technicians. 

All participants were subjected to the same locomotor training protocol, including land treadmill, underwater treadmill and kinesiotherapy exercises. Requirements for locomotor training procedures are shown in Table 2.

All dogs were evaluated at admission and at each time point, according to a neurorehabilitation checklist (Table 3), by a CCRP instructor. This checklist has key points from postural standing and sternal recumbency to bladder disfunction and pressure sores. All images were recorded with a camera (Canon EOS 2000D).

After this evaluation, two CCRP blinded observers performed the movie evaluation in slow motion and classified the dogs with the same neurorehabilitation checklist throughout this study. These two observers were blinded to each other results and to which group the dogs belong to, essential to perform an inter-agreement relation. Additionally, the two blinded observers did not speak with the CCRP instructor.

### 2.3. Outcomes

Time points of evaluations included the admission day (T0), 24 h (T1), 48 h (T2), 7 days (T3) and 15 days after admission (T4). The flow diagram describing the evolution according to the time points is illustrated in Figure 3.

### 2.4. Primary Supportive Care

All dogs were hospitalized due to the owner’s decision regarding primary care conditions, such as nutrition and hydration, that could not be done at home. The absence of a sternal recumbency ability makes it difficult for dogs to be able to eat and drink without secondary complications. Thus, all dogs rested in soft beds, with “doughnuts” bandages on the bone prominences to avoid pressure sores. Positioning was performed every 4–6 h, avoiding atelectasis and lung secretions [35]. 

Regarding nutrition, they were fed three times a day with an increase of 30% of the resting energy requirement (RER) and hydric support was performed (100–120 mL/kg/day). Some dogs had neurogenic bladders, which had to be expressed every 4–6 h as a daily rule [35,36]. All occurrences were recorded, such as pressure sores, hypoventilation, seizures and aspiration pneumonia. 

### 2.5. Locomotor Training Procedures

Participants performed the same protocol in a calm environment with music, starting with postural standing (1 min) with the help of a passive standing device (Figure 4A), followed by 30 bicycle movements in a central pad stimulation with a rough surface (Figure 4B), 4–6 times per day and 3–5 days per week. Two rehabilitation nurses and two veterinarians were needed to perform these exercises, assuring the rhythm and coordination between the forelimbs and hindlimbs, similar to locomotion. All these exercises were completed while listening to music, and their progression is described in Figure 5.

After a 30 min rest, all dogs executed the locomotor training on the land treadmill with slope (Fit Fur Life, Haslemere, UK), with the same terms regarding team members and coordination/rhythm of the movement. Training started at 0.8–1 km/h without slope and weight support by two straps, progressively increasing (Figure 6) Dogs with more than 10 kg resorted to a weight-support device (Figure 7). This was performed 4–6 times per day, 3–5 days per week, and in older dogs (≥6 kg) with monitoring by electrocardiogram (Mindray iMEC8 Vet, Shenzhen, China) and for vital parameters (e.g., heart rate, respiratory rate, perfusion parameters and rectal temperature). Dogs were excluded from this study if they had extra-systoles or other arrhythmias, and any deviations from the vital parameters were recorded daily.

The aquatic exercises were always performed during morning time and introduced carefully after 48 h of admission, especially in dogs with spinal shock signs and decreased muscle tone. To avoid suture contamination by water, technicians´ hands had to be dry and water line near the proximal epiphysis of the tibia. Water temperature was near 24–26 °C and treadmill velocity started at 1.2 km/h, from 2 to 5 min until 40 min, with progression described in Figure 8.

It is essential to mention that during the locomotor exercises, the vertebral column had to remain stable and without oscillations, avoiding other occurrences. 

Additionally, the kinematic and kinesiotherapy exercises included *cavalleti* rails, trampoline, ramps and different floor surfaces that started at day 7 (T3 time point), following the order described in Figure 9.

### 2.6. Data Collection

Data were collected from all 114 dogs and were included as categorical nominal variables: breed, chondrodystrophy, sex, neuro-location (C1–C2; C2–C3; C3–C4 and C4–C5), spinal hyperesthesia, sternal recumbency, postural standing, neurogenic bladder, neck movement, flexor reflex (thoracic and pelvic limbs), placing (thoracic and pelvic limbs), muscle tonus (extensors and flexors), pressure sores, ambulation (OFS ≥ 11) and ability to perform ≥ than 10 steps in different floors (OFS 13–14). Age and weight were the continuous quantitative variables collected. Data collection also included OFS evaluation at admission and following time points as categorical ordinal variable. Dogs with a OFS ≥ 11 were considered ambulatory and had medical discharge.

### 2.7. Statistical Analysis

Complete data were recorded in Microsoft Office Excel 365^®^ (Microsoft Corporation, Redmond, WA, USA) and processed in IBM SPSS Statistics 25^®^ (International Business Machines Corporation, Armonk, NY, USA) software. Normality tests (Shapiro–Wilk and Kolmogorov–Smirnov), arithmetic means, minimum, maximum, standard deviation (SD) and standard error of mean (SEM) were reported for continuous variables (age and weight). Descriptive statistics was performed comparing clinical and outcome variables between dogs with or without spinal hyperesthesia at admission. The two groups were compared using the non-parametric Kruskal–Wallis and Mann–Whitney tests. It was performed a univariate analysis of variance, considering OFS scores with time as a repeated measure. Chi-square tests were also performed to identify possible differences between the SHG and the NSHG.

## 3. Results

In this prospective blinded clinical study, from the 114 dogs included, 24.6% (n = 28) were mixed breed and 75.4% (n = 86) were purebred dogs, including Labrador Retriever (n = 13), Yorkshire Terrier (n = 13), Dobermann Pinscher (n = 8), Dalmatian (n = 5), Beagle (n = 4), French Bulldog (n = 4), Great Danois (n = 4), Greyhound (n = 3), Chihuahua (n = 3), Weimaraner (n = 3), Spitz (n = 3), Portuguese Podengo (n = 2), Pug (n = 2), Bouvier Bernois (n = 2), English Bulldog (n = 1), Cane Corso (n = 1), Poodle (n = 1), Epagneul Breton (n = 1), Fox Terrier (n = 1), Golden retriever (n = 1), Jack Russel terrier (n = 1), Rhodesian lion (n = 1), Briard (n = 1), Pekingese (n = 1), Pinscher (n = 1), Giant Poodle (n = 1), Alentejo Mastiff (n = 1), Rottweiler (n = 1), Soft-Coated Wheaten Terrier (n = 1), American Staffordshire Terrier (n = 1) and Dachshund (n = 1). Chondrodystrophic breeds represented 30.7% (35/114) of the study population. 

Regarding continuous variables age and weight, the total population presented normal distribution by the Kolmogorov–Smirnov test. Both the SHG and the NSHG presented similar means and medians, making groups comparable, as shown in Table 4. 

Of the total population, the most common neuro-location was at C4–C5 with total 47.4% (54/114). Additionally, in the SHG was shown a C4–C5 neuro-location with 58.8% (42/74), whereas the C3–C4 neuro-location showed to be more frequent in the NSHG with 40% (16/40). 

In regard to the neurorehabilitation check list throughout the study time points, frequency analysis of the total population was performed and is described in Figure 10 and Figure 11. 

As for the inter-observer validation, evaluations of the neurorehabilitation checklist parameters resulted in a total of 195 observations between the three observers, with only 32 inconsistencies, resulting in 17.4% inter-observer disagreement. 

The clinical sign of spinal hyperesthesia revealed 100% recovery until the end of this study, with an evident decrease during evaluation time points (Figure 12). Additionally, in the first three days after starting protocol (T2), there were no dogs in the NSHG that developed signs of hyperesthesia and there was 47.3% (35/74) in the SHG that improved from this clinical sign.

Occurrences were present in 31.58% (36/114) of the total population (Figure 13), with 31 dogs presenting pressure sores throughout the study time, 1 dog with hypoventilation and 4 dogs with aspiration pneumonia (2 of them euthanatized at T4 for further complications). No seizures were reported during the study time. Additionally, pressure sores that were present in 27.25% (31/114), revealed a solid significance between their presence and the ambulatory status achieved (X^2^(1, n = 114) = 70.304, *p* < 0.001). 

Ambulatory status was achieved in 62.3% (71/114) of total population within the first 15 days. Five dogs were ambulatory and had medical discharge at time point T2, followed by 14 dogs and 52 dogs at the time points T3 and T4, correspondingly (Figure 14). Considering the ambulatory dogs, 32.4% (23/71) showed the ability to perform ≥ than 10 steps in different floors (OFS 13 or 14), by the time of medical discharge. Of the total population, 43 dogs remained non-ambulatory until T4 (day 15).

The SHG was able to achieve ambulatory status in 48.6% (36/74) of dogs and 17.6% (13/74) achieved ability to perform ≥ than 10 steps in different floors, whereas in the NSHG, ambulatory status was reached in 87.5% (35/40) of dogs and 25% (10/40) had the ability to perform ≥ than 10 steps in different floors. Comparing the two groups, a significant difference concerning their ability to achieve ambulation status was observed (X^2^ (1, n = 114) = 16.683, *p* < 0.001).

Regarding the dogs that achieved ambulatory status, when considering the SHG and the NSHG, a strong significance between spinal hyperesthesia at admission and the number of days until ambulation was achieved (X^2^ (2, n = 71) =10.329, *p* = 0.006).

OFS scores obtained at the time of medical discharge were compared in both the SHG and the NSHG using non-parametric the Kruskal–Wallis test and the non-parametric Mann–Whitney test, which revealed evident difference in the distribution of OFS scores between groups (*p* < 0.001). Additionally, according to the univariate analysis of variance, a significant difference between both groups in each time point was observed (*p* < 0.001; *adjusted R^2^* = 0.809). OFS scores for the SHG and the NSHG were registered and estimated marginal means evolution chart is shown in Figure 15.

## 4. Discussion

This prospective blinded clinical study included 144 dogs with cervical compressive myelopathy, who were subjected to a first neurorehabilitation consultation, selecting only 114 dogs with C1–C5 neuro-location compatible with chondroid metaplasia, thus IVDD—Hansen type I. According to the literature [11], this specific neuro-location is most seen in small breed dogs. However, in the present study, the main frequency was the Labrador Retriever with 11.4% (13/114) with the Dobermann Pinscher (n = 8) as the third most registered breed, possibly due to the higher frequency of Hansen type I and the relative vertebral canal stenosis [37,38]. 

Our study had only 30.7% (35/114) of chondrodystrophic breeds, which was not in accordance to previous studies [11]. The mean age was 7.78 years old in the SHG, similar to the NSHG (mean age 8.07), with a median of 8 in both groups, assuring homogeneity between them. The presence of IVDD associated with increase in age [37] may be a possible reason for this result. In regard to weight, there was a normal distribution, making both groups comparable. 

Cervical compressive myelopathy or cervical spondylomyelopathy do not have a simple pathophysiology; however, it seems to be related to static and dynamic factors [39,40,41]. In the present study, dogs were interpreted with a static element and compressive spinal cord disease that was approached by a single-ventral slot decompressive surgery, agreeing with da Costa (2010) [41], that previously reported a long-term success of nearly 72%, and other authors [16,20,42]. 

To Bonelli and colleagues (2021) [43], 80% of the dogs with a single affected site were younger than 6 years of age, while most dogs with multiple sites affected were 6 years or older, not identical to our study based on middle-aged to older dogs subjected to a single-ventral slot approach [44,45,46]. 

In previous studies [1,47,48], the C2–C3 and C3–C4 intervertebral sites are the most common injury locations, in contradiction to our population with the highest frequency at C4–C5 in ~60% at the SHG and ~30% at the NSHG.

The division of the study population into the SHG and the NSHG was based on the hyperesthesia clinical sign. Both groups underwent the same neurorehabilitation procedures, including hospital care, nutrition and environment adaptation. The admission to the rehabilitation centers until 3 days after surgery and the observation of primary neurological signs (e.g., spinal shock), defined an essential timeline detail of this study design, needed to show if early locomotor training could potentially increase pain or neurological deficits. 

The hyperesthesia secondary to cervical compressive myelopathy may be caused by compression of the spinal cord if disc material extrusion occurs in a dorsolateral direction (e.g., between the dorsal longitudinal ligament and vertebral venus sinus), by root nerve compression, by meningeal irritation [2], or by damage in soft tissue and ligamentum flavum, mainly present in dogs with multiple compression sites and foraminal stenosis [49]. 

In the present study, with a single ventral slot approach, the possibility of pain with progressive persistency or new cases of hyperesthesia in the NSHG, could indicate that early prescription of locomotor training was not a possibility in post-surgical dogs. However, in Figure 10 and Figure 12, the frequency analysis of this clinical sign on the total population, revealed an exponential decrease in each time point, with no dog manifesting pain within two weeks. In Figure 10, it can be observed that normal micturition was recovered mostly in T4 (day 15) in 45% (43/95) of the dogs.

The study design resorted to a neurorehabilitation checklist to help in the evaluations at each time points (Figure 10 and Figure 11), such as the sternal recumbency ability, which increased from no dog to 80% (76/95) at T4, and the postural standing ability that achieved 66% (63/95). 

As mentioned above, spinal hyperesthesia revealed 100% of resolution until the end of this study, without resorting to the usual rehabilitation modalities for neuropathic and nociceptive pain treatment (e.g., transcutaneous electrical stimulation, interferential electrical stimulation, laser therapy, and electroacupuncture) [50]. The introduction of these modalities would result in a potential bias for the study design, although they could be useful for reorganization of the descending tracts, neurogenesis and strengthening the pre-existing neural tracts both cranially and caudally to the injury, and possibly through it, promoting the anatomic and synaptic plasticity [51,52,53,54]. Multidisciplinary protocols are based in functional electrical stimulation and transpinal stimulation [53,55], which could also be implemented in tetraplegic patients who suffered injury in the phrenic motoneuron pools and/or phrenic nerves to improve ventilatory status [56,57]. Although in our study, dogs were only under locomotor training to improve ambulation.

In Figure 12, regarding spinal hyperesthesia throughout the different time points, it is possible to observe that even with the implementation of early locomotor training at T2 (day 3), 47.3% (35/74) dogs improved from that clinical sign in the SHG and no dog developed new signs of hyperesthesia in the NSHG, suggesting that locomotor training could be introduced in the first days after surgery, contrary to the usual cage rest recommendation or simply restricted activity using a body harness and analgesia [27]. Additionally, da Costa (2010) [41] has reported that physical therapy should be considered as a treatment for dogs with severe cervical myelopathies and Zidan et al. (2018) [58] suggested that early post-operative rehabilitation was safe in dogs with incomplete SCI after surgery for thoracolumbar IVDD. 

Some authors [59,60] reported that dogs bear 60% of their body weight in the forelimbs and nearly 40% on the hindlimbs, which is an important consideration when implementing weight-supported training and encouraging correct postural standing ability. That is the main reason for dogs with more than 10 kg resorted to a weight-supported device, considering our population of tetraplegic dogs with a weight mean of 22.07 kg (SHG) and 18.23 kg (NSHG).

The BWSTT, in both human and veterinary medicine, has already proved that allows a repetitive strep training for several consecutive days, starting on T0 (day of admission) to promote early increase in muscle force output and endurance with rhythmic activity by stimulating the spinal locomotor circuitry and remnants ascending/descending pathways [28,61,62,63]. In addition, to Frank and Roynard (2018) [50], locomotor training itself could be helpful in decreasing pain conditions, such as allodynia.

The tetraplegic dogs needed strict and complex care, resting in soft beds with doughnuts bandages at bone prominences to avoid occurrences such as pressure scores [3,64]. In this study, dogs had acute (2–48 h after injury) or sub-acute (48 h to 14 days) presentation [65,66], thus without pressure sores. However, 27.2% (31/114) developed this occurrence, revealing a solid significance between their presence and the ambulatory status achieved (*p* < 0.001). Additionally, from the 36 dogs who manifested any type of occurrences, 11% (4/36) developed aspiration pneumonia, resulting in the euthanasia of two dogs. However, it is important to mention that early implementation of locomotor training was not directly associated with any of these occurrences. Therefore, a relation between complications and decreasing neurological status or even hemodynamic instability, as observed in other studies, was not observed [67]. 

This study was conducted with ethical guidelines to understand if this approach was a safe and beneficial treatment, resulting in 62.3% (71/114) of ambulatory status (OFS ≥ 11) within 15 days, from the total population (Figure 14). Of these 71 dogs, 32.4% (23/71) manifested the ability to perform more than 10 steps in different floor types (OFS 13 and 14) by the time of medical discharge.

Furthermore, comparing both groups, a significant difference in their ability to achieve ambulatory status was observed (*p* < 0.001) and in the NSHG 25% (10/40) achieved a OFS ≥ 13–14. Additionally, there was a clear difference between spinal hyperesthesia at admission and the number of days until ambulation (*p* < 0.006). These results suggested that hyperesthesia could be a clinical sign related to inflammation, oedema and compression of the neural tissues. To support the comparison between the SHG and the NSHG, it was used a non-parametric test that confirmed the significant difference between them (*p* < 0.001). This difference was also observed in each time point (*p* < 0.001) with a R^2^ = 0.809, as shown in Figure 15. For example, in T2, the OFS-estimated marginal mean was 8.8 in the NSHG and 7.1 in the SHG.

Inter-observer evaluations had a disagreement < 20% between the CCRP blinded observers, which indicates that early locomotor training could be a safe tool to be implemented in the first 3 days after surgery in tetraplegic dogs with cervical IVDD—Hansen type I. Additionally, allowing the observation that the clinical sign of hyperesthesia should be considered in the recovery of these dogs.

This clinical study proposed new research tools that lead us to understand and justify the rupture of the cage rest or reduced exercise activity paradigm in these populations of dogs, suggesting early and intensive training [28,29,32,58,68,69,70,71], now applied in cervical C1–C5 compressive myelopathies. In contrast to the usual physical rehabilitation that is recommended with cryotherapy, passive range of motion, massage, assisted standing and walking [29,72,73,74,75,76]. Although in acute ataxic dogs, cage rest could be essential to reduce pain and inflammation secondary to affected nerve roots and meninges [77,78]. 

A new perspective is needed based on a multidisciplinary team approach, with knowledge in the rehabilitation field and equipped with essential devices to assure correct postural standing (Table 2). All the rehabilitation procedures allowed the synaptogenesis of rubrospinal and medullary reticulospinal tracts, facilitating the flexor muscles and inhibiting the extensors [79,80]. The kinesiotherapy exercises promoted the imbalance correction, while the vestibulospinal tract stimulation played an essential role in posture [54,80]. The propriospinal fibers also should be stimulated by multisynaptic pathways, re-establishing the connections between descending motor tracts, the central pattern generators, intrinsic circuitry and interneurons located on the pelvic and thoracic intumescences [81,82,83,84]. All those exercises were performed in an integrated sensory environment (e.g., listening to music), promoting somatosensory stimulation, as mentioned by Lewis and collaborators (2022) [72].

One of this study’s limitations is the number of dogs that should be higher. Additionally, the absence of long-term follow-ups to understand the evolution of the 43 dogs that did not achieve ambulatory status until day 15, and the absence of a detailed comparison between each parameter of the neurorehabilitation checklist after T4 in those dogs, were also limitations that should be considered. 

## 5. Conclusions

Early locomotor training in tetraplegic post-surgical dogs with acute/subacute cervical IVDD—Hansen type I following a single ventral slot approach may be a possible and safe treatment to be implemented within the first 3 days after surgery, allowing 62.3% (71/114) of recovery within 15 days of protocol, with no occurrences registered. Therefore, there was no relation between complications or decreased neurological status and early locomotor training implementation.

Furthermore, spinal hyperesthesia was an important clinical sign to consider and throughout the different time points, it was possible to observe that even with the implementation of early locomotor training at T2 (day 3), 47.3% (35/74) dogs improved from that clinical sign in the SHG and no dog developed new signs of hyperesthesia in the NSHG, suggesting that locomotor training could be introduced in the first days after surgery, contrary to the usual cage rest recommendation. Thus, dogs manifesting hyperesthesia may need more time until recovery, although the locomotor training performed did not increase pain or other neurological deficits. 

## Figures and Tables

**Figure 1 animals-12-02369-f001:**
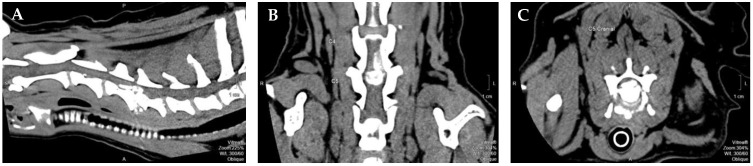
Computed tomography-myelogram image from a 9-year-old mixed breed dog with a single compressive disc extrusion in C4–C5. (**A**) Sagittal view; (**B**) dorsal view; (**C**) transverse view. Figure courtesy of Professor Dr. António Ferreira from the Neurology department of Veterinary Medicine Faculty, University of Lisbon, Portugal. Legend: P–Posterior; S–Superior; A–Anterior.

**Figure 2 animals-12-02369-f002:**
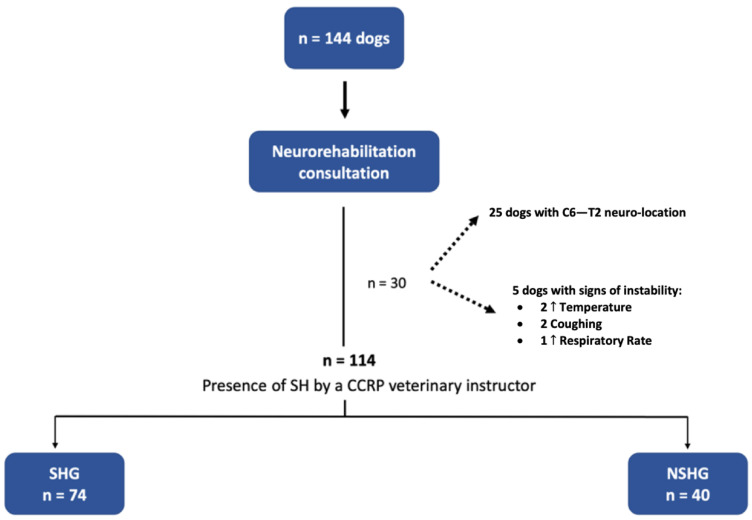
Study flow diagram. SH—spinal hyperesthesia; CCRP—Certified Canine Rehabilitation Practitioner; SHG—spinal hyperesthesia group; NSHG—non-spinal hyperesthesia group.

**Figure 3 animals-12-02369-f003:**
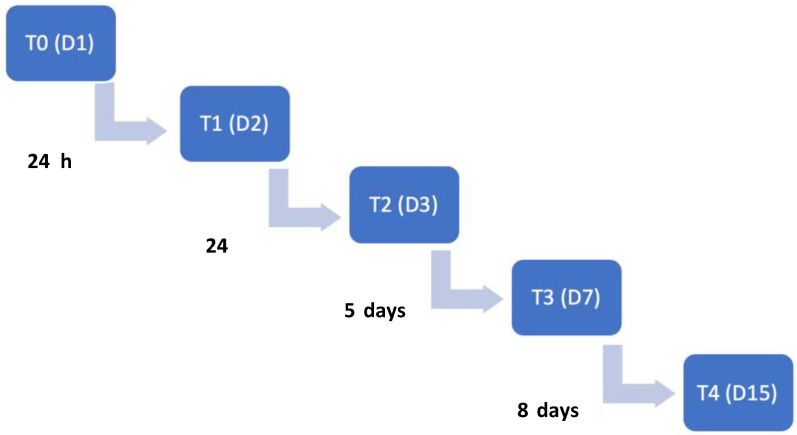
Time points flow diagram throughout this study. T0: admission day; T1: day 2; T2: day 3; T3: day 7; T4: day 15. T: time point; D: day.

**Figure 4 animals-12-02369-f004:**
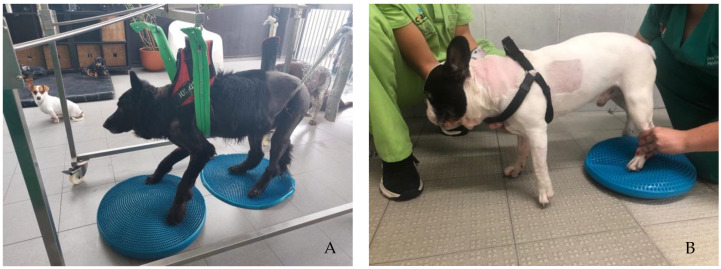
(**A**) Postural standing exercise with the help of a passive standing device; (**B**) bicycle movements in a central pad stimulation with a rough surface.

**Figure 5 animals-12-02369-f005:**
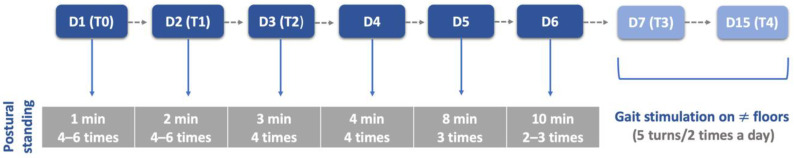
Postural standing exercises protocol throughout this study. D: day; T: time point; T0: admission day; T1: day 2; T2: day 3; T3: day 7; T4: day 15; ≠: different.

**Figure 6 animals-12-02369-f006:**
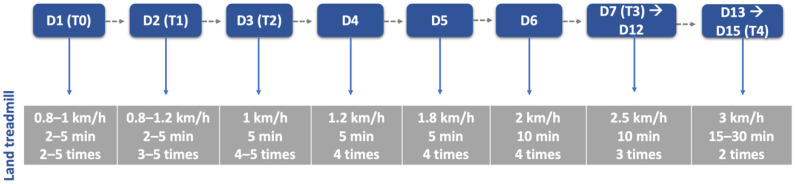
Land treadmill protocol throughout this study. D: day; T: time point; T0: admission day; T1: day 2; T2: day 3; T3: day 7; T4: day 15.

**Figure 7 animals-12-02369-f007:**
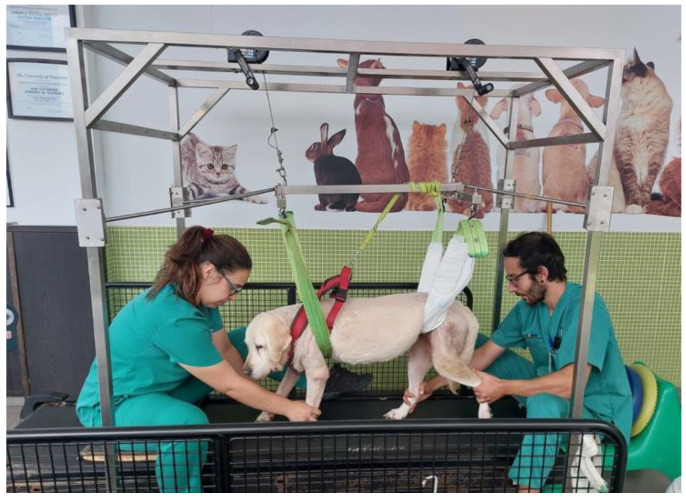
Locomotor training resorted to a weight-support device for dogs with more than 10 kg.

**Figure 8 animals-12-02369-f008:**
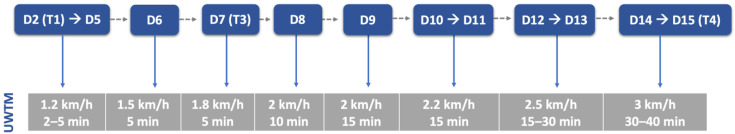
Underwater treadmill exercises throughout this study. UWTM: underwater treadmill; D: day; T: time point. T1: day 2; T3: day 7; T4: day 15.

**Figure 9 animals-12-02369-f009:**
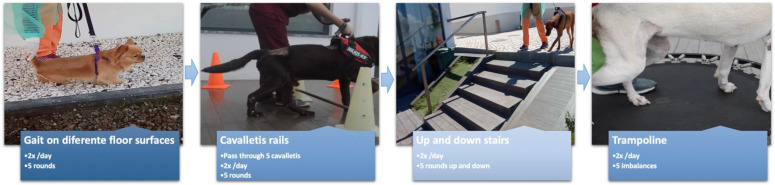
Kinesiotherapy circuit performed by this order, between day 7 and day 15. Patients performed one circuit in the morning and the other in the afternoon. If non-ambulatory patients, two rehabilitation technicians were needed and support was made with a harness (forelimbs) and by the tail or with straps (hindlimbs).

**Figure 10 animals-12-02369-f010:**
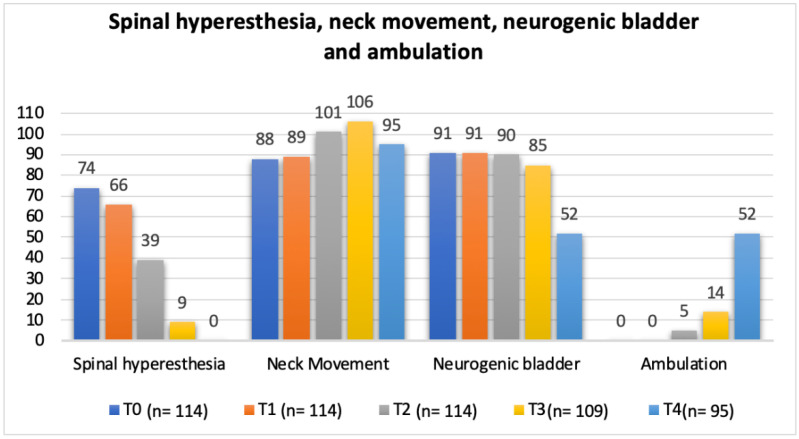
Total population frequency analysis of spinal hyperesthesia, neck movement, neurogenic bladder and ambulation evolution throughout the study time points. T0: admission day; T1: day 2; T2: day 3; T3: day 7; and T4: day 15; x-axis: neurorehabilitation check list parameters; y-axis: frequency number.

**Figure 11 animals-12-02369-f011:**
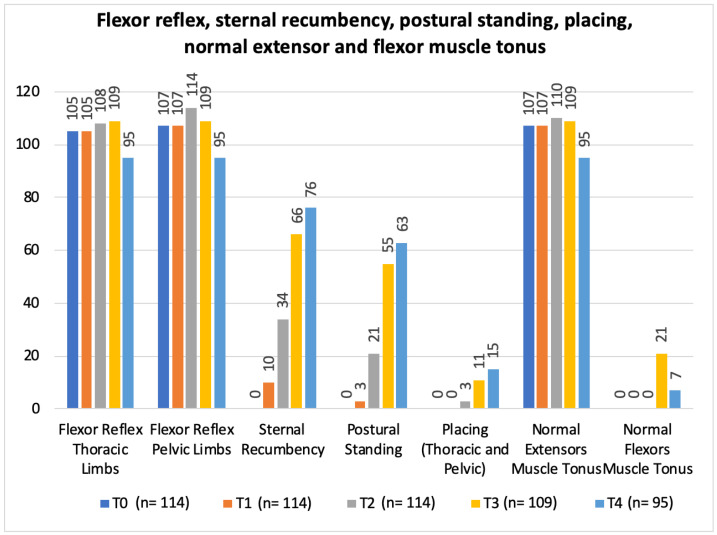
Total population frequency analysis of flexor reflex, sternal recumbency, postural standing, placing, normal extensor and flexor muscle tonus evolution along the study time points. T0: admission day; T1: day 2; T2: day 3; T3: day 7; and T4: day 15; x-axis: neurorehabilitation check list parameters; y-axis: frequency number.

**Figure 12 animals-12-02369-f012:**
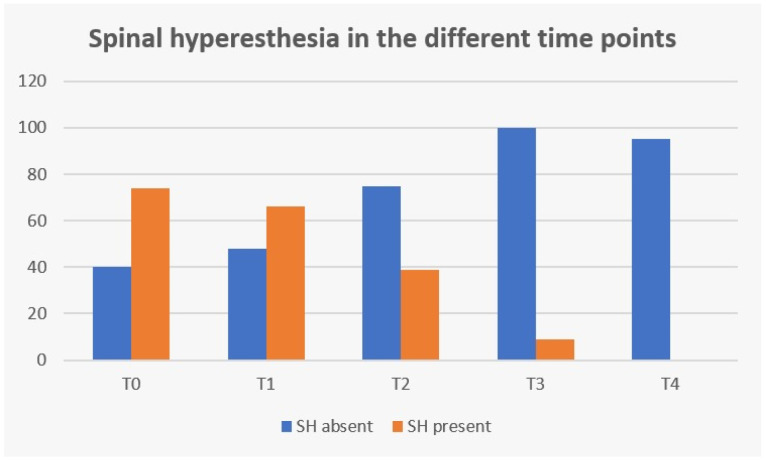
Frequency analysis of spinal hyperesthesia (SH) during study time points. T0: admission day; T1: day 2; T2: day 3; T3: day 7; and T4: day 15; x-axis: time points of this study; y-axis: frequency number.

**Figure 13 animals-12-02369-f013:**
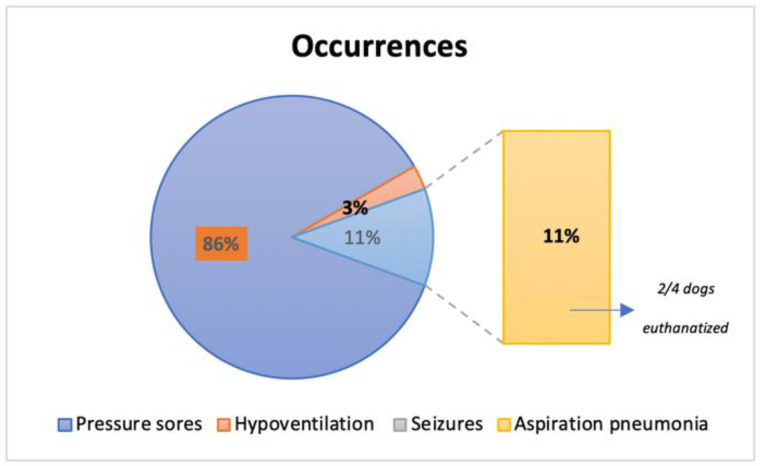
Frequency analysis of occurrences reported in the 36 dogs throughout this study.

**Figure 14 animals-12-02369-f014:**
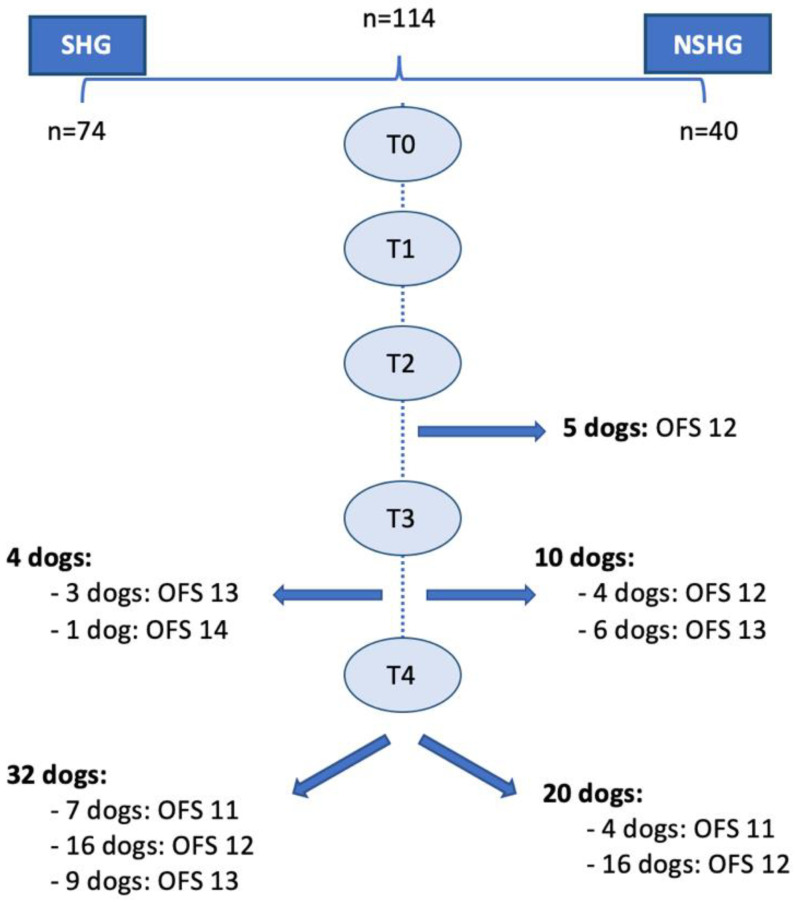
Diagram of the ambulation status and medical discharge. SHG: spinal hyperesthesia group; NSHG: non-spinal hyperesthesia group; OFS: open field score; T0: admission day; T1: day 2; T2: day 3; T3: day 7; and T4: day 15.

**Figure 15 animals-12-02369-f015:**
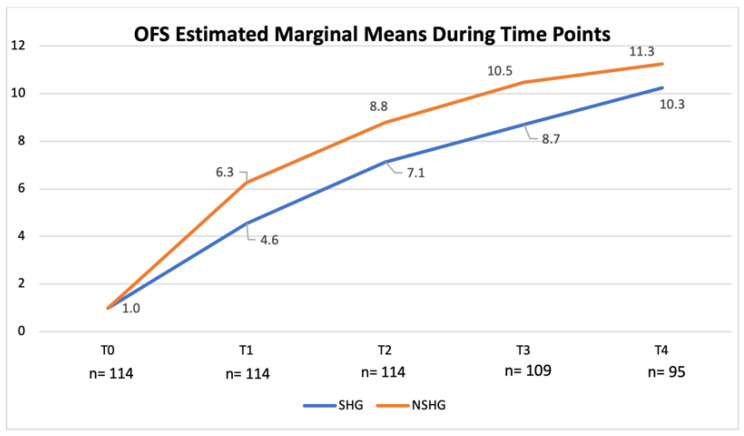
Evolution of the open field score (OFS)-estimated marginal means in the spinal hyperesthesia group (SHG) and the non-spinal hyperesthesia group (NSHG), throughout the study time points. T0: admission day; T1: day 2; T2: day 3; T3: day 7; and T4: day 15; x-axis: time points of this study; y-axis: OFS.

**Table 1 animals-12-02369-t001:** Characterization of the total population.

Characterization Parameters	SHG (n = 74)	NSHG (n = 40)	Total (n = 114)
Breed	Breed: 63/74 (85.1%) Mixed breed: 11/74 (14.9%)	Breed: 23/40 (57.5%) Mixed breed: 17/40 (42.5%)	Breed: 86/114 (75.4%) Mixed breed: 28/114 (24.6%)
Chondrodystrophy	Present: 29/74 (39.2%) Absent: 45/74 (60.8%)	Present: 6/40 (15%) Absent: 34/40 (85%)	Present: 35/114 (30.7%) Absent: 79/114 (69.3%)
Sex	Male: 52/74 (70.3%) Female: 22/74 (29.7%)	Male: 17/40 (42.5%) Female: 23/40 (57.5%)	Male: 69/114 (60.5%) Female: 45/114 (39.5%)
Age	<7 years: 24/74 (32.4%) ≥7 years: 50/74 (67.6%) Mean: 7.78 years	<7 years: 17/40 (42.5%) ≥7 years: 23/40 (57.5%) Mean: 8.07 years	<7 years: 41/114 (36%) ≥7 years: 73/114 (64%) Mean: 7.89 years
Weight	≤10 kg: 24/74 (32.4%) >10 kg: 50/74 (67.6%) Mean: 22.07 kg	≤10 kg: 7/40 (17.5%) >10 kg: 33/40 (82.5%) Mean: 18.23 kg	≤10 kg: 31/114 (27.2%) >10 kg: 83/114 (72.8%) Mean: 20.72 kg
Neuro-location	C1–C2: 8/74 (10.8%) C2–C3: 9/74 (12.2%) C3–C4: 15/74 (20.3%) C4–C5: 42/74 (56.8%)	C2–C3: 12/40 (30%) C3–C4: 16/40 (40%) C4–C5: 12/40 (30%)	C1–C2: 8/114 (7%) C2–C3: 21/114 (18.4%) C3–C4: 31/114 (27.2%) C4–C5: 54/114 (47.4%)
Spinal hyperesthesia	Present: 74/74 (100%)	Absent: 40/40 (100%)	Absent: 40/114 (35.1%) Present: 74/114 (64.9%)
Sternal recumbency	Absent: 74/74 (100%)	Absent: 40/40 (100%)	Absent: 114/114 (100%)
Postural standing			
Neurogenic bladder	Absent: 17/74 (23%) Present: 57/74 (77%)	Absent: 6/40 (15%) Present: 34/40 (85%)	Absent: 23/114 (20.2%) Present: 91/114 (79.8%)
Neck movement	Absent: 26/74 (35.1%) Present: 48/74 (64.9%)	Present: 40/40 (100%)	Absent: 26/114 (22.8%) Present: 88/114 (77.2%)
Flexor reflex (forelimbs)	Present: 68/74 (91.9%) Absent: 6/74 (8.1%)	Present: 37/40 (92.5%) Absent: 3/40 (7.5%)	Present: 105/114 (92.1%) Absent: 9/114 (7.9%)
Flexor reflex (hindlimbs)	Present: 68/74 (91.9%) Absent: 6/74 (8.1%)	Present: 39/40 (97.5%) Absent: 1/40 (2.5%)	Present: 107/114 (93.9%) Absent: 7/114 (6.1%)
Ambulation	Absent: 74/74 (100%)	Absent: 40/40 (100%)	Absent: 114/114 (100%)
Placing (fore and hindlimbs)			
Normal flexor muscle group tonus	Abnormal: 74/74 (100%)	Abnormal: 40/40 (100%)	Abnormal: 114/114 (100%)
Normal extensor muscle group tonus	Normal: 68/74 (91.9%) Abnormal: 6/74 (8.1%)	Normal: 39/40 (97.5%) Abnormal: 1/40 (2.5%)	Normal: 107/114 (93.9%) Abnormal: 7/114 (6.1%)
Pressure sores	Absent: 74/74 (100%)	Absent: 40/40 (100%)	Absent: 114/114 (100%)

Legend: SHG—spinal hyperesthesia group; NSHG—non-spinal hyperesthesia group.

**Table 2 animals-12-02369-t002:** Key points—requirements for locomotor training procedures.

Material	Land treadmill; Underwater treadmill; Passive standing device; Harness; Four support straps; Cavaletti rail; Trampoline; Ramps; Different floor surfaces.
Rehabilitation Team	Dogs ≤ 10 kg: One technician (Two in the first week); Dogs > 10 kg: Two technicians (Four in the first week).

**Table 3 animals-12-02369-t003:** Key Points—neurorehabilitation checklist.

Time Points	T0	T1	T2	T3	T4
Postural standing (30 s)					
Absence of spinal hyperesthesia (palpation of the vertebral column from C1–T2)					
Normal neck movement					
Normal flexor reflex (forelimbs and hindlimbs)					
Normal placing test (forelimbs and hindlimbs)					
Ability to perform until 10 steps without falling (OFS ≥ 11)					
Ability to perform ≥ 10 steps in different floors (OFS 13/14)					
Normal muscle tone (palpation of the extensor muscles)					
Normal muscle tone (palpation of the flexor muscles)					
Sternal recumbency					
OFS evaluation					
Pressure sores					
Neurogenic bladder					

**Table 4 animals-12-02369-t004:** Descriptive analysis of age and weight in both the spinal hyperesthesia group and the non-spinal hyperesthesia group.

		SHG (n = 74)	NSHG (n = 40)	Total (n = 114)
Age (years)	Mean	7.78	8.07	7.89
	Median	8	8	8
	Variance	11.514	14.533	12.474
	SD	3.393	3.812	3.532
	Minimum	1	1	1
	Maximum	14	16	16
	SEM	0.394	0.603	0.331
	Normality Test	0.009 (Kolmogorov–Smirnov)	0.587 (Shapiro–Wilk)	0.015 (Kolmogorov–Smirnov)
Weight (kg)	Mean	22.07	18.23	20.72
	Median	21.5	19	20.5
	Variance	183.571	139.82	170.239
	SD	13.549	11.825	13.048
	Minimum	3	3	3
	Maximum	62	45	62
	SEM	1.575	1.87	1.222
	Normality Test	0.066 (Kolmogorov–Smirnov)	0.019 (Shapiro–Wilk)	0.033 (Kolmogorov–Smirnov)

Legend: SHG—spinal hyperesthesia group; NSHG—non-spinal hyperesthesia group.

## Data Availability

The data presented in this study are available upon request from the corresponding author.

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
