# Peer review of "Early Locomotor Training in Tetraplegic Post-Surgical Dogs with Cervical Intervertebral Disc Disease"

_animals, 2022, doi:10.3390/ani12182369_

Round 1

Reviewer 1 Report

Dear Authors

This is a well-written article with solid methodological background, proper examples were chosen and considered. My minor comments concern the Conclusions section, which in my opinion should be completed.

Regards

Author Response

Dear Authors

This is a well-written article with solid methodological background, proper examples were chosen and considered. My minor comments concern the Conclusions section, which in my opinion should be completed.

Thank you so much for your comment and your appreciation. As suggested we completed the conclusion section.

Reviewer 2 Report

First of all, I would like to highlight the interest in the article as well as the high number of clinical cases found over the 5 years (July 2017 to July 2022). Although in [473] they indicate that the number of cases may be a limitation, I do not agree, given the difficulties in having a casuistry like the one included in the study. It is even more valuable that they were able to conduct this research without outside funding. My congratulations.

On the other hand, I completely agree with what is described in [404-405]. This article is good evidence that dogs do not need to be in a cage after surgery. In future research, it might be advisable to compare the results obtained after using locomotor training with respect to the use of other physical therapies. If they foresee the use of radiofrequency, I offer to collaborate with the authors.

Author numbering: review.

In the Abstract perhaps they can already indicate the meaning of SHG (Spinal Hyperesthesia Group) and NSHG (No Spinal Hyperesthesia Group)

[6]: Problems in reading Arrábida and Setúbal (check source)

[58]: space after [15]

[127]: Change “table” to “Table” (first letter capitalized, if we follow the same procedure as in the rest of the text)

Table 1: change “anos” to “years”

Table 2: assess removing the vignette in each sentence

[152] Change “Requerimentos” to “Requirements”

[153] Change “table” to “Table” (first letter capitalized, if we follow the same procedure as in the rest of the text)

When they talk about “breed dogs”… do they mean purebred dogs?

[258]: “Grand Danois” or “Great Danois” ?

[268] Change “table” to “Table” (first letter capitalized, following the same procedure as in the rest of the text)

[288-290]: check if they have used another source. In the pdf it is not well appreciated, but it seems that it is a different font than the rest of the text.

Author Response

General comments:

First of all, I would like to highlight the interest in the article as well as the high number of clinical cases found over the 5 years (July 2017 to July 2022). Although in [473] they indicate that the number of cases may be a limitation, I do not agree, given the difficulties in having a casuistry like the one included in the study. It is even more valuable that they were able to conduct this research without outside funding. My congratulations.

Thank you so much for your comment and your appreciation. I agree with you, however for some readers that are not from this field and in terms of statistics, some may consider this a limitation and that is our reason for writing this.

On the other hand, I completely agree with what is described in [404-405]. This article is good evidence that dogs do not need to be in a cage after surgery. In future research, it might be advisable to compare the results obtained after using locomotor training with respect to the use of other physical therapies. If they foresee the use of radiofrequency, I offer to collaborate with the authors.

Thank you so much for your comment, I would really like to work with you further in this project, maybe after this is finished we can make contact.

Specific comments:

Author numbering: review.

- Thank you so much for your comment, as suggested we corrected and reviewed the author numbering.

In the Abstract perhaps they can already indicate the meaning of SHG (Spinal Hyperesthesia Group) and NSHG (No Spinal Hyperesthesia Group)

- As suggested we added the meaning for SHG and NSHG in the abstract.

[6]: Problems in reading Arrábida and Setúbal (check source)

- Arrábida and Setúbal are Portuguese names of Portugal locations.

[58]: space after [15]

- Thank you for your comment, space was added.

[127]: Change “table” to “Table” (first letter capitalized, if we follow the same procedure as in the rest of the text)

- As suggested by the reviewer we replaced table for Table.

Table 1: change “anos” to “years”

- Thank you so much for your comment and correction, it was a mistake of translation. It is already replaced.

Table 2: assess removing the vignette in each sentence

- As suggested we removed the vignette.

[152] Change “Requerimentos” to “Requirements”

- Thank you so much for your comment and correction, it was a mistake of translation. It is already replaced.

[153] Change “table” to “Table” (first letter capitalized, if we follow the same procedure as in the rest of the text)

- As suggested by the reviewer we replaced table for Table.

When they talk about “breed dogs”… do they mean purebred dogs?

- Yes that is it. We replaced breed dogs for purebred dogs.

[258]: “Grand Danois” or “Great Danois” ?

- We change Grand for Great.

[268] Change “table” to “Table” (first letter capitalized, following the same procedure as in the rest of the text)

- As suggested by the reviewer we replaced table for Table.

[288-290]: check if they have used another source. In the pdf it is not well appreciated, but it seems that it is a different font than the rest of the text.

- Thank you so much for this comments, we checked the font.